# Anticipation of sexually arousing visual event leads to overestimation of elapsed time

**Ville Johannes Harjunen**[1]*, **Michiel Spapé**[2], **Niklas Ravaja**[1]

**1** Faculty of Medicine, Department of Psychology and Logopedics, University of Helsinki, Helsinki, Finland,
**2** Centre for Cognitive and Brain Sciences, University of Macau, Macau, China

* ville.harjunen@helsinki.fi

**Data Availability Statement:** Data files and analysis scripts are available at the Open Science Framework repository (https://osf.io/7t9jx/).

**Funding:** This work was supported by the VIRTUALTIMES (Horizon 2020, Contract No 824128; Authors funded by the project: V.J.H.; M.

## Abstract

Subjective estimates of duration are affected by emotional expectations about the future. For example, temporal intervals preceding a threatening event such as an electric shock are estimated as longer than intervals preceding a non-threatening event. However, it has not been unequivocally shown that such temporal overestimation occurs also when anticipating a similarly arousing but appealing event. In this study, we examined how anticipation of visual erotic material influenced perceived duration. Participants did a temporal bisection task, where they estimated durations of visual cues relative to previously learned short and long standard durations. The color of the to-be-timed visual cue signalled either a chance of seeing a preferred erotic picture at the end of the interval or certainty of seeing a neutral grey bar instead. The results showed that anticipating an appealing event increased the likelihood of estimating the cue duration as long as compared to the anticipation of a grey bar. Further analyses showed that this temporal overestimation effect was stronger for those who rated the anticipated erotic pictures as more sexually arousing. The results thus indicate that anticipation of appealing events has a similar dilating effect on perceived duration as does the anticipation of aversive events.

## Introduction

Perception of time is an essential part of human experience and behavior [1,2]. Motor actions, speech, and nonverbal communication are all dependent on accurate duration estimation [3,4]. Yet, the human experience of time is relative changing according to expectations, sensory feedback, motor actions, and experienced emotions [5–7]. A good example of this relativity is that people perceive emotionally arousing events as longer lasting than neutral, non-arousing events of the same duration [8–11]. Recent research has shown that this emotion-driven overestimation of elapsed time occurs not only when exposed to an emotional event but also when expecting something emotionally salient to occur [12,13]. Understanding how emotional expectations about sensory events modify time perception is crucial for discovering the temporal principles that guide human perception and cognition [14] as well as the interindividual variation in processes such as reward anticipation [15]. So far, the research on temporal distortions driven by emotional anticipation has focused on anticipation of threats such as

S.; N.R.) funded as a part of the Future and Emerging Technologies (FET) Project (FETPROACT-01-2018: Emerging paradigms and communities) of European Union. The open access fee will be paid by Helsinki University Library. The funders had no role in study design, data collection and analysis, decision to publish, or preparation of the manuscript.

**Competing interests:** The authors have declared that no competing interests exist.

aversive images, electric shocks, or loud noises [12,16,17]. However, it has remained unclear whether anticipation of positively valenced and appealing events likewise dilates estimated duration.

Direct exposure to threatening pictures of angry faces and aversive scenes has been shown to result in overestimation of duration [5,18–21]. Similarly, just expecting an aversive event to occur is shown to lengthen estimated time [11,13,16]. Whether direct exposure to positively valenced events or their anticipation likewise results in temporal overestimation remains unclear due to mixed evidence. For example, in some studies images of positive content (smiling faces, food, people, pets) were estimated to last longer than images of neutral content [19,21,22]. In other studies, no such effect was found [23]. A study looking into emotional anticipation found that only the anticipation of negative (e.g., surgery) but not positive (e.g., babies, pets) visual stimuli resulted in temporal overestimation when compared to neutral visual stimuli [24]. It is thus possible that either anticipation of positively valenced events does not cause temporal overestimation [13] or that to cause a temporal dilation the anticipated positive event needs to be perceived as intrinsically rewarding and arousing.

While direct empirical evidence is lacking when it comes to the link between anticipation of appealing events and temporal dilation, research on reward anticipation suggests that such a causal link might exist. For example, in a study where macaque monkeys were conditioned to expected liquid reward after certain visual stimuli found that the activity in their midbrain dopaminergic neurons increased linearly with the logarithm of the waiting time [25]. In human neuroimaging studies it has been shown that prospects of reward speed up reactions in a timing task and that a prefrontal-striatal circuit that processes the accumulating temporal information is also involved in processing the reward prospects [26]. Indeed, converging evidence from neuroimaging and lesion studies suggest that the nigrostriatal dopaminergic pathway of the dorsal striatum of the basal ganglia play a crucial role in time perception [27]. Therefore, anticipation of reward influences duration estimation via modulating a common striatal dopaminergic mechanism [28].

The temporal distortions resulting from both direct exposure to and anticipation of negative and positive emotional events is often explained in terms of the so-called pacemaker-accumulator model [29] extended in the Scalar Expectancy Theory [30,31]. According to this model, emotional events or their anticipation increases the perceiver's arousal level which speeds up an internal pacemaker mechanism resulting in faster accumulation of temporal information (i.e. pulses). The perceiver uses the number of accumulated pulses as a measure of elapsed time so that higher pulse count results in greater estimates of elapsed time [19,32]. Consequently, time intervals accompanied by heightened emotional arousal are estimated as lasting longer. Therefore, according to the pacemaker accumulator framework, anticipating appealing events should likewise lead to temporal dilation.

To test the hypothesis that anticipation of an appealing event induces overestimation of elapsed time, we conducted a behavioral study where participants classified durations of visual cues relative to long and short standard durations. The cue color indicated whether an erotic image could be presented at the end of the interval. We chose erotic pictures matched with participants' sexual preferences as a proxy of appealing events because these visual stimuli are commonly marked with positively valenced high-arousal emotional state [33] and because these stimuli are associated with rapid activation of the dopaminergic reward system [34]. The cue color indicated whether there was a chance of a subsequent presentation of an erotic picture. As this chance did not always materialise, it enabled us to differentiate between three experimental conditions: 1) the erotic image was anticipated and presented at the end of the estimated time interval; 2) the erotic image was anticipated but a grey bar was shown; and 3) a grey bar was anticipated and presented.

The benefits of including the second condition (a picture was anticipated but grey bar was shown) were three-fold: First, it allowed us to disentangle the effects of anticipation of appealing events from the effects of direct exposure to the appealing events. That is, if the anticipated event would always follow the anticipatory cue [11,12], it would be impossible to know whether the distortion was really due to the anticipation or due to the event itself. Second, inclusion of the condition with anticipation followed by a grey bar made the anticipatory cue condition and the neutral cue conditions comparable in terms of their perceptual features. This is an important measure of control since the difference in perceptual features between the contrasted stimulus conditions can have a confounding effect on duration estimation [35]. Finally, uncertainty of the appealing event's occurrence is likely to increase emotional arousal resulting from the anticipation. This assumption is supported by the findings showing that people are more motivated to seek for uncertain rewards than certain rewards and experience the uncertainty as emotionally arousing [36]. Due to the three aforementioned factors, we were primarily interested in the difference between the neutral cue condition and the anticipatory cue condition followed by a neutral visual event.

Our first hypothesis (H1) was that people estimate cue durations as lasting longer when the cue color signals a change of a subsequent appealing event than when it signalises a neutral visual stimulus. After testing the first hypothesis, we examined whether the anticipation-related temporal dilation could be explained by the level of arousal associated with the anticipated event. We hypothesized (H2) that the level of sexual arousal reported during anticipation should be positively related to the anticipation-related temporal overestimation.

## Materials and methods

### Participants

The sample of 46 healthy adult volunteers was recruited via the University of Helsinki student organisation email lists. No exclusion criteria were used for the participation but those who self-assessed having a strong negative reaction to explicit erotic visual content were not encouraged to participate in the study. The participation was anonymous and the experimental system did not collect any identifying information from the participants. Required sample was estimated based on expected difference in duration estimates between picture anticipation and neutral (grey bar) condition. Previous studies on anticipation of aversive events showed effect sizes varying between medium ($d$ = .76; [16,37]) and large ($d$ = 1.89; [12]). Therefore, we expected the effect size to be of medium or large size (e.g., $d$ = .75). Power analysis based on expected effect of $d$ = .75 (https://jakewestfall.shinyapps.io/pangea/) informed that finding such effect with 80% statistical power and 0.05 alpha level required a minimal sample of 39 participants. Therefore, we aimed at collecting a sample of N ≥ 40. Out of the 46 participants, three participants reported all kinds of erotic pictures as not sexually arousing. The observations of these three individuals were excluded from the sample because no appealing picture category could be defined. One participant was found to give monotonic responses in the timing task (evaluating all stimuli as "long") and the data was therefore excluded. The final sample of 42 individuals with an average age of 26.02 (SD = 6.52) years consisted of 29 females, 15 males, and two who selected "other" or did not report their gender.

### Ethics statement

The University of Helsinki Research Ethics Committee in the Humanities and Social and Behavioural Sciences reviewed the study plan at the Board meeting on September 13, 2023. The review board found that based on the received material the planned study followed the ethical principles of research in the humanities and social and behavioral sciences issued by

the Finnish Advisory Board on Research Integrity. Thus, the review board stated that the mentioned study was ethically acceptable. Participants were informed of their right to withdraw from the study at any point and the anonymity of their responses. Written informed consent was given by the participants in the online experiment system by "ticking" a box ("With this selection I confirm my consent described above and take part in the study as a volunteer participant"). The experiment was terminated and no data was collected if the box was not selected prior pressing ENTER to continue. Therefore, the data was recorded only from those who gave their consent. A seven euros compensation was paid after completion of the experiment to those who provided a completion certification code via email. This code could not be linked to the research data of individual subjects.

## Procedure

The data were collected remotely using E-prime Go, a self-contained run-time version of the E-Prime 3 [38]. The experiment was run locally by the participants who downloaded the experiment via a link attached to the invitation letter. The experiment was hosted on an online server and run locally on the participants' PC computer from which the data was sent to the server after completion of the task. Before starting the task, the participants were informed about the experiment and their right to withdraw from the study at any point. After giving informed consent, they were asked about their personal preferences concerning explicit erotic visual content (see Measures). These ratings were used to compose a stimulus set that was in alignment with the preferences of each participant. There was no option of homosexual material because the used picture database did not include pictures of homosexual interaction. The participants rated each category of erotic content on a five-point Likert scale (1: not at all arousing; 5: very arousing). A set of 20 images were added to the personalised stimulus from the highest rated picture category. However, if all the category-specific arousal ratings were lower than 3, the preferred category was not defined and a random selection from all categories was used instead. In a later stage, the data of individuals with no preferred category were removed due to the absence of appealing anticipated events. The experimental session consisted of three phases: conditioning, training, and bisection task phase.

**Conditioning.**   After giving informed consent and indicating their preferred picture category, participants went through a short conditioning phase to learn to associate a certain colored (e.g., pink) cross cue to a subsequent presentation of an erotic picture and another colored cross cue (e.g., blue) to a presentation of a grey bar. The pictures shown during the conditioning phase all represented the personally preferred picture category. Each trial began with a central fixation dot of 900–1400 ms (randomized), followed by a cross cue of 3000 ms. Depending on the cue color, an erotic picture or a grey bar (non-threat trial) was presented remaining in view for 500 ms. The condition session consisted of eight trials, four with erotic picture and four with a grey bar. In case an erotic picture was shown, the participants evaluated how sexually arousing they found the picture (see Measures).

**Training.**   In the training phase, the participants learned to distinguish between a short duration (800 ms) and a long duration (2600 ms). The training consisted of 14 trials with seven trials of short (800 ms) cue duration and seven trials of long (2600 ms) cue duration, all presented in a random order. Each trial initiated with a fixation dot shown for 900–1400 ms (randomized) after which a cross cue was shown. The particpants had to estimate the presentation duration of this cue. In the training phase, the cue color was always indicating a neutral condition (grey bar) and the grey bar was shown in each trial after the cue disappeared. In the first four trials, after the grey bar was shown for 500 ms, the participants were explicitly told whether the cue duration they just perceived was long or short and they were asked to press

either the "z" (for short) or "m" (for long) letter key of their keyboard to indicate the correct response (key-response mapping counterbalanced between participants). In the remaining 10 training trials, the correct answer was not shown but participants had to indicate whether they perceived the cue more as short or long by pressing either the "z" or "m" key. After each response, the participants received feedback about whether their response was correct. Following the feedback, a 1200–1600 ms blank inter-trial interval (ITI) followed.

**Bisection task.** Following the training phase, participants went through the bisection task where their task was to evaluate whether a presented cue duration was more similar to the short or the long duration learned in the training phase. This time, the cue duration was varied among seven durations (800, 1100, 1400, 1700, 2000, 2300, or 2600 ms) instead of two (800 ms and 2600 ms) and the cue condition was varied between anticipation of an erotic picture and anticipation of a grey bar by varying the cue color. The task consisted of 448 trials which were divided into four blocks of 112 trials in each. Each trial started with a 900–1400 ms (randomized) fixation followed by a colored cross cue (see Fig 1). The cue color indicated whether there was a chance of a subsequent erotic picture to be shown after the cross disappeared or whether a grey bar would be shown instead. After the cue disappeared, the erotic picture or the grey bar was presented for 500 ms. In half of the trials (i.e., 224), the cue indicated that a grey bar would be presented at the end, and the grey bar was also shown (i.e., neutral condition). In the other 224 trials, the cue indicated a chance of an erotic picture. However, only in half (112) of these anticipation trials the picture was actually shown (i.e., anticipation+picture) and in the other half of the trials the grey bar was shown (i.e., anticipation+blank). After the grey bar/ erotic picture disappeared, the participants indicated using the z and m keys (key-response mapping counterbalanced between participants) whether the cue duration was more similar to the short or the long duration learned in the training phase. Thereafter, a 1200–1600 ms blank inter-trial interval (ITI) followed. Altogether, for each comparison duration (800, 1100, 1400, 1700, 2000, 2300, and 2600 ms), there were 32 neutral cue trials, 16 anticipation+picture trials, and 16 anticipation+blank trials.

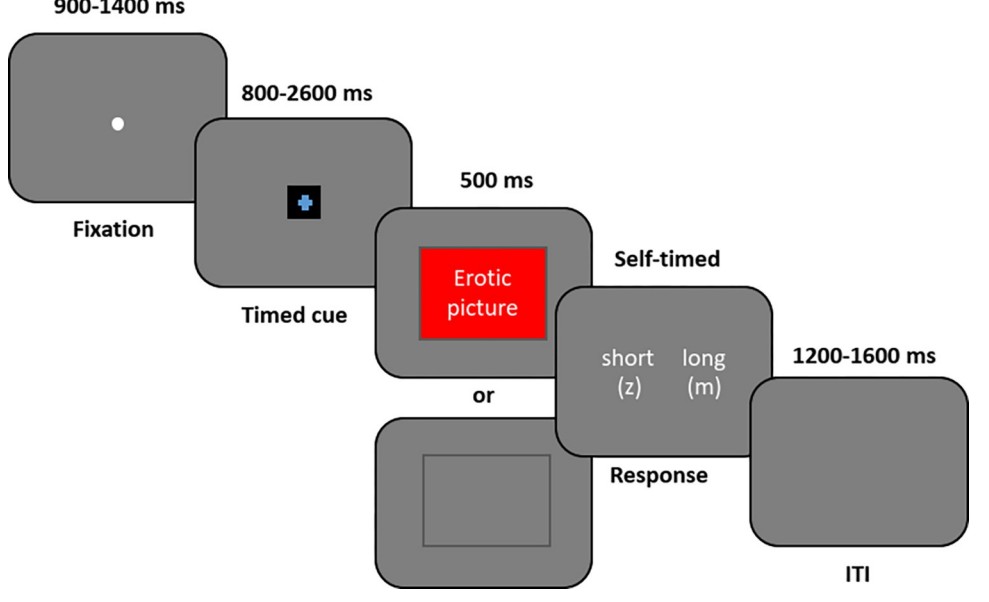

**Fig 1. Trial procedure with timing information.**

## Stimuli

The experiment was developed using E-prime 3.0 software (Psychology Software Tools Inc, Sharpsburg, PA) and converted to the E-prime Go system for online data acquisition. The E-prime Go experiment was hosted on an online server and could be run on the participant's own Windows PC. Upon task completion, the data were saved to the online server. Due to the remote data collection approach, the display technology and response devices varied across participants. To ensure the devices met the minimal requirements of E-Prime 3.0, we examined the Experiment Advisor Reports (EARs) generated by E-prime Go.

**Visual stimuli.** The fixation was a white dot with a font size of 18 presented on a grey background in the middle of the screen. The cue of which duration the participants were asked to estimate consisted of a black square (28 x 28 pixels) and a colored cross (i.e., plus sign with font size 30) within it. The cross colors were equiluminant light pink (HSV coordinates: 308, 75, 100) and light blue (HSV: 211, 75, 100) of which intensity and saturation were held constant and only the hue was varied. Erotic picture stimuli consisted of 60 images (see S1 Appendix for picture numbers) drawn from the International Affective Picture System (IAPS, [39]. Only images depicting naked bodies, sexual poses, or sexual behavior were included. Based on a technical report of the IAPS [39], these erotic pictures are rated on 9-point rating scale as positively valenced (pictures of couples: M = 6.61, SD = 0.46, pictures of women: M = 5,97, SD = 0.61, pictures of men: M = 5.61, SD = 0.46) and moderately arousing (pictures of couples: M = 6.44, SD = 0.36, pictures of women: M = 5.61, SD = 0.58, pictures of men: M = 4.81, SD = 0,54). The grey bar shown in neutral condition was grey (RGB: 127, 127, 127) with hue of 160, luminance of 120 and zero saturation.

## Measures

**Personal preferences in erotic visual material.** The preference ratings concerned visual erotic content of nude men (How sexually arousing do you find erotic images of men?), nude women (How sexually arousing do you find erotic images of women?), and heterosexual couples (How sexually arousing do you find erotic images of heterosexual couples?). The ratings were made on a five-point Likert scale (1: Not at all arousing; 5: Very arousing). Pictures belonging to the category rated as most sexually arousing were used in the conditioning and bisection task phases of the experiment.

**Picture-related sexual arousal.** Following presentation of an erotic picture in the conditioning phase, participants were asked to rate sexual arousal related to the presented pictures ("How sexually arousing was the image") on a five-point Likert scale (1: not at all arousing; 5: very arousing). In the conditioning phase, four pictures (randomly selected from the preferred category) were presented and rated. The average person-specific arousal score across the four ratings was calculated for each individual after which the scores were standardised using the Z-score method.

**Duration estimation.** First, the proportions of "long"-responses in each comparison duration were calculated separately for each participant and each anticipation condition. Then, a sigmoidal S-shaped mathematical function was fitted to the cleaned binary response (long vs. short) data using responses the "long"-responses as outcome and cue duration as the predictor. The function was fitted separately to the data of each participant and each cue condition using estimation by direct maximization of the likelihood [40,41]. It followed the formula:

$$\psi(x) = \gamma + (1 - \gamma - \lambda) \cdot fun(x),$$

Where $\gamma$ refers to guess rate, $\lambda$ to lapse rate ja *fun* to sigmoidal function, with asymptotics $y = 0$ and $y = 1$. $\lambda$ was set to 0.015 and $\gamma$ was set to 0.

The resulting psychometric curves were used to extract three Bisection points (BPs) for each participant, each corresponding to a different cue condition. BPs represented perceived duration of comparison intervals at the 50% probability of "long" response. These BP values were used in the statistical testing as the outcome variable.

## Data analysis

First, we examined the proportion of preferred picture categories as indicating the variety of utilized erotic pictures. Then, picture-related sexual arousal ratings obtained from the conditioning phase were examined to ensure the participants perceived the pictures as sexually arousing. Here, one-way independent samples ANOVA was used to test differences in rated arousal among the three picture categories. In regard to the bisection task data, "long/short" responses with too fast or slow reaction times were first removed from the data before the analysis using the median absolute deviation (MAD) method with a conservative outlier threshold value of 3 [42]. MAD was used instead of commonly applied Z-score method due to the strongly right-skewed distribution of the RT values. Then, psychometric functions were fitted using the quickpsy command of the quickpsy R package [43] (version: 0.1.5.1) to the cleaned binary trial-level data to extract bisection point (BP) values. We hypothetized the average BP of the anticipation+blank condition to be smaller as compared to the BP of neutral cue trials, indicating over-estimation of elapsed time in the context of picture anticipation. While the hypothesis was limited to the two cue conditions with comparable perceptual features, the initial comparison of conditions included all three conditions (i.e., also the anticipation+picture condition).

To examine differences in average BPs among the three conditions, we calculated a multi-level linear model (MLM) using the lmer function for R. The restricted maximum likelihood (REML) method was used for model estimation. MLM was used instead of conventional repeated measures ANOVA as allowing estimation of cross-level interaction effects between the cue condition and person-specific average sexual arousal scores in the later stage of analysis. Testing such cross-level interactions between repeated measures factors and person-level covariates with repeated measures ANCOVA violates the ANCOVA's assumption of homogenous regression slopes [44]. Therefore, MLM, which allows regression slopes to vary between conditions, was better suited for the statistical testing. The initial MLM contained a fixed effect of cue condition with three levels in the factor (neutral, anticipation+blank, and anticipation+-picture). In the follow-up analyses, person-level sexual arousal score and the interaction effect between cue condition and sexual arousal score were added to the model as fixed effects. The interaction effect was examined to see whether people who found the pictures more sexually arousing exhibited stronger differences in their bisection points between the anticipation +blank and neutral cue condition.

Finally, to see whether there were differences in the temporal sensitivity among the three conditions, Weber fraction (WF) was calculated separately for each condition and each participant. The function followed the formula:

$$WF = \frac{t[p(Long = .75)] - t[p(Long = 0.25)]}{2 \cdot t[p(Long = .50)]}$$

Where t refers to the stimulus duration that corresponds to p(Long) on the psychometric curve.

In duration estimation, the WF refers to the just noticeable difference in temporal duration of a presented stimulus [40]. The lower the ratio, the more sensitive the estimation is for

detecting differences in the presented duration. To test the difference in WFs among the three cue conditions, a separate MLM was calculated setting WF as dependent variable and cue condition as the fixed main effect.

In all models predicting BP and WF, the intercept was defined as a random effect. No random slopes were included as adding the random slope of cue condition to the model would have made the random-effects parameters and the residual variance to become unidentifiable. The omnibus test of the fixed effects utilised type-III analysis of variance with Satterthwaite's method. Planned pairwise comparisons of estimated margin means were calculated as post-hoc analyses of the main effect of cue condition. The interaction effect between arousal score and cue condition was probed using simple slopes.

## Results

### Picture preferences and sexual arousal ratings

While the average arousal related to pictures of women was higher as compared to the other two picture categories (see Table 1), the differences among the three image categories did not reach statistical significance, $F(2, 39) = 1.27$, $p = .293$.

Likewise, male participants did not differ statistically significantly in their picture-related arousal ($M = 3.67$, $SD = 1.16$) from the female participants ($M = 3.71$, $SD = 1.22$), $t(30.56) = 0.12$, $p = .908$. When examining participants' preferences related to the three picture categories, we found that the majority (N = 21) found erotic images of heterosexual couples as the most arousing category. Eleven participants found erotic images of women the most arousing and ten participants rated erotic images of men as the most arousing.

### Effect of event anticipation on duration estimation

The analysis of duration estimation data was started by first scanning the response durations using the median absolute deviation (MAD) method [42] to detect trials, which differed substantially from each subject's reaction time (RT) distribution. Trials of which RTs exceeded the conservative threshold value of 3 were defined as outliers and removed from the data resulting in 9.12% reduction of the total number of observations. Then, a sigmoidal S-shaped mathematical function was fitted to the cleaned binary response (long vs. short) data using responses as the outcome and cue duration as the predictor. The function was fitted separately to the data of each participant and each anticipation condition.

Fig 2 shows the average probabilities of "long" responses as a function of seven comparison durations (dots) in the three cue conditions. The curves represent the fitted psychometric functions and the vertical lines crossing the curves at 50 percent probability indicates the bisection points (BP). As can be seen, the BPs of both anticipation conditions occur at shorter durations than the BP of the neutral condition. This indicates that the probability of estimating the

**Table 1.** Stimulus-related sexual arousal ratings, proportions of female and male participants, and sample sizes of different picture preference groups.

| Preferred picture category | Sexual arousal rating | | Gender distribution | | Total N |
|---|---|---|---|---|---|
| | *M* | *SD* | **Females** | **Males** | |
| Pictures of heterosexual couples | 3.42 | 1.06 | 13 | 8 | 21 |
| Pictures of women | 4.11 | 1.02 | 5 | 6 | 11 |
| Pictures of men | 3.70 | 1.54 | 8 | 1 | 10 |

*Note*. One participant who reported their gender as other preferred pictures of men.

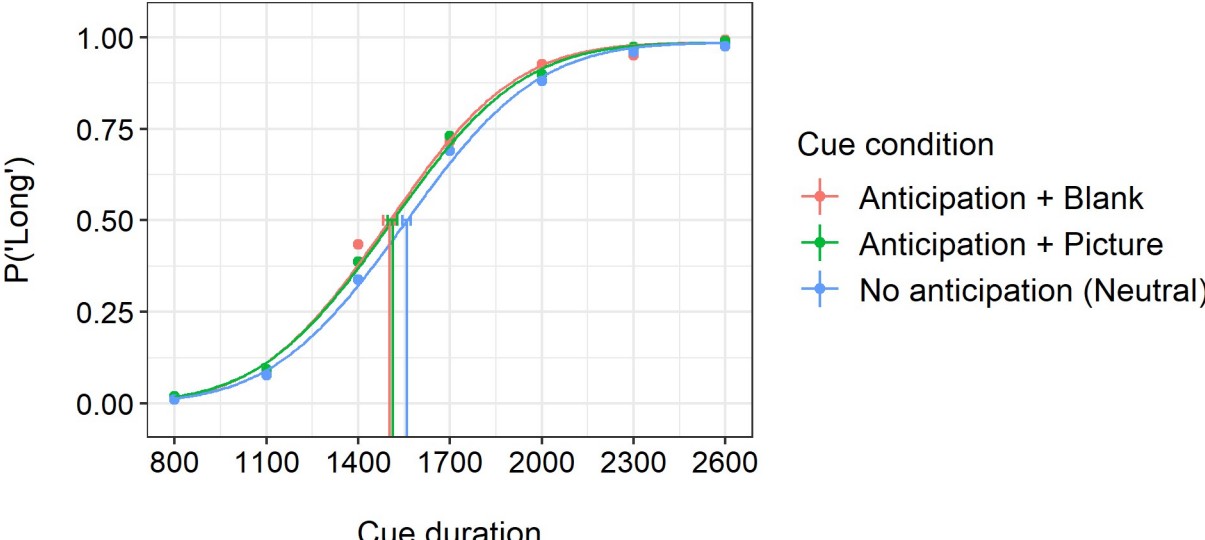

**Fig 2. Probabilities of cues perceived as long in the neutral (no anticipation), anticipation+blank, and anticipation+picture condition as a function of cue duration.** The colored dots represent average probabilities of the "long" responses in each cue duration and each cue condition.The three curves present psychometric functions fitted to the binary bisection task responses. The vertical lines crossing the curves represent 50% bisection points and their error bars refer to 95% confidence intervals obtained with a parametric bootstrap method.

duration as long was higher when anticipating an appealing event than when the event was not anticipated. Examining the psychometric functions separately for each individual revealed substantial interindividual variation in the BPs particularly in the anticipation+picture condition. In approximately 1.5% of trials with the longest cue duration (2600 ms), people erroneously estimated the duration as short, for which reason a fixed 1.5% lapse rate was utilized in the fitting of the psychometric functions.

Next, MLM was conducted to test H1 according to which anticipation of an appealing event would result in higher probability of "long" responses than anticipating a neutral event. The model included the fixed effect of cue condition. The omnibus F-test of cue-condition reached statistical significance, $F(2, 82) = 3.58$, $p = .030$. Estimated margin means of the condition-specific BPs are shown in Fig 3 along with the "raw" BP values. As can be seen, average BP for the anticipation+blank ($M = 1509.89$, $SE = 34.40$) was lower than average BP for the neutral condition ($M = 1568.341$, $SE = 34.40$), $t(82) = 2.35$, $p = .021$, $d = .26$. The same was the case between the anticipation+picture ($M = 1511.83$, $SE = 34.40$) and the neural condition, $t(82) = 2.28$, $p = .026$, $d = .25$. However, in the anticipation+picture condition, a single very low BP value was observed. Removal of the value caused the difference between the anticipation+picture and neutral condition to become non-significant, $t(81.1) = 1.91$, $p = .060$, $d = .21$.

### Moderating effect of sexual arousal on anticipation-related temporal overestimation

To test H2 and examine whether self-reported sexual arousal ratings of the pictures explained variation in the temporal overestimation effect observed between the anticipation+blank and neutral condition, we conducted a follow-up model where the individuals' picture-related arousal ratings were entered to the MLM model as a person-level moderator. Both fixed main effects of the cue condition (anticipation+blank vs. neutral) and the arousal score as well as

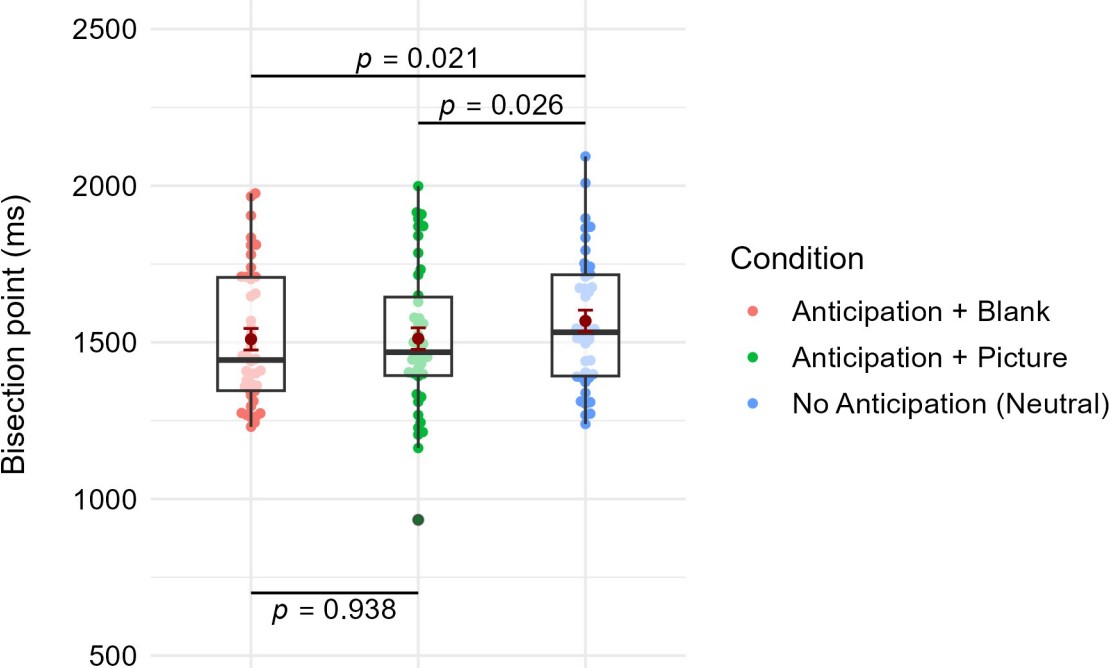

**Fig 3. Comparison of the BPs among the three cue conditions.** Dark red dots and their error bars refer to MLM-based estimated margin means and their standard errors, respectively. P-values refer to planned contrasts between the estimated margin means. Colored dots refer to the "raw" BP values used in the MLM model as predicted outcome. Overlying boxplot Figs show the medians (black horizontal line) as well as lower and upper quartiles and whiskers of the distributions.

their cross-level interaction effect were estimated. Omnibus F-tests of the model effects revealed a significant main effect of cue condition, $F(1, 40) = 11.61$, $p = .002$, reflecting the already observed overestimation effect due to anticipation. Moreover, while the main effect of arousal was non-significant, $F(1, 40) = 0.58$, $p = .449$, the interaction between arousal and cue condition was statistically significant, $F(1, 40) = 5.39$, $p = .026$. The model coefficients for the fixed effects implied that the association between BP and arousal was more negative in the anticipation +blank than in the neutral condition, $b = -40.05$, $SE = 17.25$, $t(40) = -2.32$, $p = .026$, $d = .37$. Simple slopes calculated based on the model coefficients confirmed this and showed that individuals who found the pictures more sexually arousing had lower BPs and therefore higher probabilities in estimating the duration as long in the anticipation+blank than in the neutral cue condition (Fig 4). The results thus indicated that rating the pictures more sexually arousing is associated with stronger temporal overestimation when anticipating the content to be shown.

### Effect of event anticipation on temporal sensitivity

To examine differences in the temporal sensitivity among the three cue conditions, a MLM was calculated to predict WF setting the cue condition as a predictor. Differences in WF among the conditions were statistically significant, $F(2, 82) = 3.55$, $p = .033$). Average WF of the anticipation+picture condition ($M = .11$, $SD = .05$) was lower than in the anticipation +blank ($M = .12$, $SD = .05$), or in the neutral condition ($M = .13$, $SD = .04$). Pairwise comparisons of the estimated margin means of WFs revealed that only the anticipation+picture differed significantly from the neutral condition, $t(82) = -2.648$, $p = .026$, $d = .29$, while the other contrasts were non-significant ($ps > .26$). This indicates that the sensitivity was higher in the anticipation+picture condition where the anticipated event was actually presented.

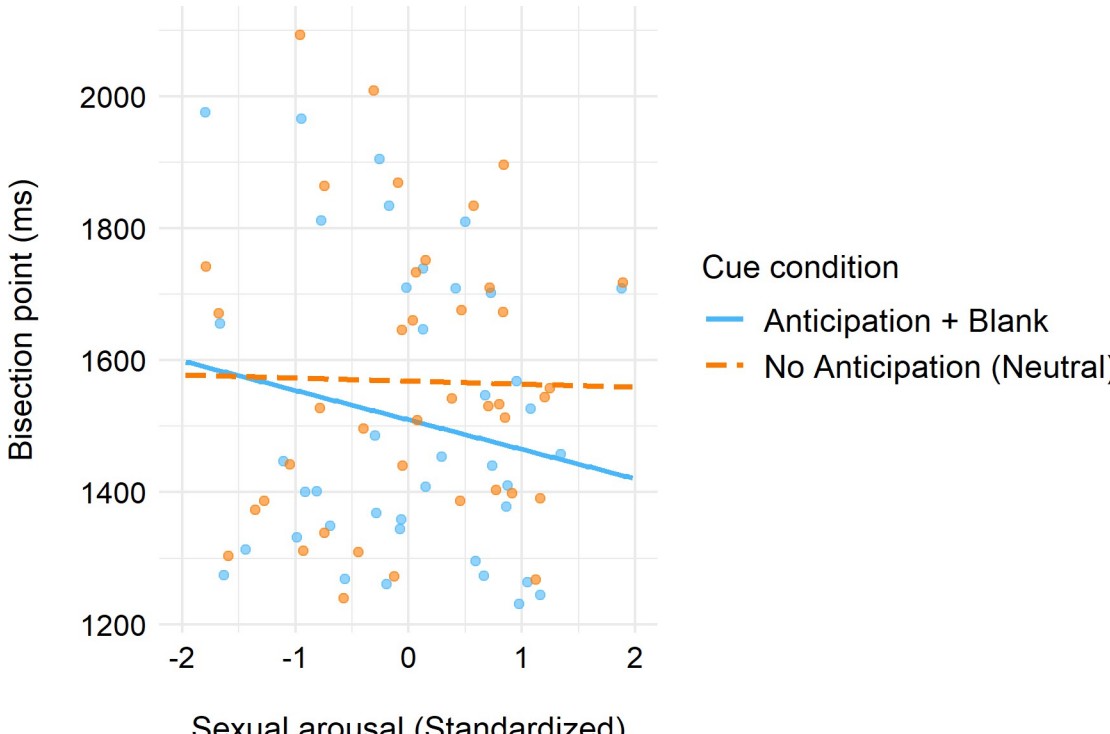

**Fig 4. Simple slopes showing the linear relationships between sexual arousal scores and estimated temporal bisection point in two cue conditions: Neutral and anticipation+blank.** The lines represent regression slopes calculated based on the MLM coefficients of fixed effects. Sexual arousal was standardised around the grand mean.

## Discussion

In this study, we examined whether anticipation of a sexually appealing visual event as compared to a neutral visual event causes overestimation of elapsed time (H1) and whether this overestimation can be explained by sexual arousal elicited by the anticipated event (H2). Supporting H1, we found that anticipating an appealing event leads to an overestimation of the elapsed time. This overestimation effect was found to be similar regardless of whether the anticipated event was shown in the end. However, follow-up analyses after outlier removal revealed the effect being more robust in the condition where a picture was anticipated but not shown. Giving support for the second hypothesis, we found that those who found the erotic pictures as more sexually arousing exhibited greater temporal overestimation due to the anticipation of picture presentation. In this section, we will elaborate the findings in detail in light of previous related research.

Our first finding that expecting a sexually appealing picture to be presented on screen increases the likelihood of temporal overestimation is consistent with previous laboratory studies where direct exposure to positively valenced visual stimuli (e.g., pictures of smiling faces) has been found to be perceived lasting longer than direct exposure to emotionally neutral visual stimuli [19,21,22]. However, it is important to note that most of the previous studies used non-erotic positive stimuli. An early study by Angrilli and colleagues [18], which used the same erotic IAPS stimuli as did we found that direct exposure to erotic pictures resulted in an underestimation of elapsed time. While the current finding seems opposite to the findings by Angrilli and colleagues [18], the discrepancy can well be due to different timing tasks used in the two studies (reproduction vs. temporal bisection task). Indeed, emotional conditions

that produce overestimation in temporal bisection task may not similarly influence temporal reproduction [45]. Nevertheless, our findings are in alignment with previous research related to temporal overestimation caused by anticipation of negative emotional events [11,12,16]. As these studies utilized the same timing task and operated on the same sub-second time scale, their results can be considered more informative for understanding the current findings. In these studies, on threat anticipation, the observed overestimation effect was attributed to an increase in arousal that was suggested to accelerate an internal pacemaker leading to higher number of pulses being accumulated over a time [19]. The current findings suggest that similar acceleration of pulse rate occurs also when expecting an appealing sexual event.

Recent studies on emotion-driven temporal distortions have demonstrated caveats to the arousal-based account of temporal dilation showing, for example, that direct exposure or expectation of emotional events can also shorten the perceived duration [37,46]. The emotion-driven underestimation found in these studies was suggested to be due to the reallocation of attentional resources away from the timing task to the emotional procession of the event itself [13,37]. Indeed, according to the attentional gate model of duration estimation, shifting attention away from the timing results in less pulses being accumulated, and duration being estimated as shorter [47]. Sarigiannidis and colleagues [37] suggested that the attentional shift may happen more likely when the occurrence of the anticipated emotional event is uncertain. In a recent study, we examined whether people exhibit temporal underestimation when anticipating an uncertain visual threat that was aversive, but not seen as dangerous as electric shocks used by Sarigiannidis and colleagues [37]. We found that anticipating an uncertain visual threat elicited temporal overestimation rather than underestimation. Therefore, we argued that the direction of temporal distortion depends not only on uncertainty but on the level of danger associated with the uncertain event. The current study results show that anticipation of uncertain appealing events also causes temporal overestimation rather than underestimation. It is thus possible that the arousal resulting from emotional anticipation boosts the accumulation of temporal evidence and causes temporal dilation as long as the anticipation and preparation for the emotional becomes prioritized and starts taxing the attentional resources allocated to the timing task.

In the current study, we also examined the differences in temporal sensitivity. In many previous studies no differences have been found in the temporal sensitivity between neutral and emotional conditions (Droit-Volet & Gil, 2009). However, we found a slight difference between the neutral and anticipation condition that was followed by the picture presentation. The sensitivity was found to be slightly higher in the picture condition. Since the picture was shown prior to the "long"/" short" response, the result suggests that seeing the picture influenced not the so-called clock stage involving the pacemaker-accumulator device but the decision-making and memory stages of duration estimation [31]. Seeing the picture might have, for example, motivated the participants to put more effort into the memory-based comparison of durations.

Although the sets of appealing visual stimuli were matched to the participants' sexual preferences, participants were found to vary substantially in how sexually arousing they found the personalized picture sets. Interestingly, these differences in sexual arousal ratings predicted how strongly the participants overestimated the picture-signalling cue durations. The finding speaks for great interindividual variation in anticipation-related temporal distortions but it is also in alignment with the pacemaker accumulator model that presents arousal as the key mechanism behind temporal dilation [5]. Previous research on sexual dysfunctions, such as premature ejaculation (PE) disorder, has shown that men with inability to delay ejaculation during intercourses how also faster reactions when anticipating erotic images and reduced hedonic valuation of non-sexual rewards [48]. The current findings imply that the greater hedonic valuation of sexual rewards results also to stronger temporal dilation. Whether the

temporal distortions caused by anticipated sexual rewards are stronger in people with sexual dysfunctions and addictions should be investigated in the future.

Of course, the arousal measure used in the current study also had its limits as it was not applied during the actual bisection task but during the condition phase of the experiment. Therefore, the acquired ratings might be more informative about the participant's general tendency to get aroused by erotic pictures of their choice than about their acute sexual arousal during the anticipation period. Therefore, more direct electrophysiological measures of arousal should be utilized in future to examine further the association between arousal and temporal dilation. Finally, it is important to better differentiate general autonomic arousal from sexual arousal since while the anticipation of erotic content likely increases autonomic arousal, it does not necessarily initiate the cycle of sexual arousal.

Another limitation that could be taken into account in future research is that in the current study the anticipation conditions were only contrasted to a condition in which a grey blank bar was shown. While by doing so equalized the low-level perceptual features between the neutral and anticipation condition, one could still argue that the difference in estimated duration was due to the mere expectation. Indeed, expecting even a soft tone leads to a longer estimated time than expecting nothing to happen (Droit-Volet et al., 2010). However, it is unlikely our observation of temporal overestimation was due to the expectation per se because the effect was moderated by ratings of sexual arousal.

Finally, it is important to note that the sample size was underpowered for a reliable detection of small person-situation interaction effects. The requires sample size was estimated based on the main effect of cue condition that was the focus of our main analysis. Therefore, the observed small-to-medium (d = .37) interaction effect between cue condition and arousal may have been too small to be detected reliably with 80% statistical power when increasing the alpha level to the conventional 5% threshold. Therefore, further replication of the results is needed to ensure robustness of the reported findings.

To conclude, the current study demonstrated that anticipating uncertain appealing visual events results in temporal dilation and that sexual arousal related to the appealing events predicts stronger temporal dilation. The results speak for the centrality of arousal in the distortions of subjective time and suggest that not only the anticipation of threat, but anticipation of rewards can cause subjective time to slow down.

## Supporting information

**S1 Appendix. List of utilized IAPS pictures.**
(DOCX)

## Acknowledgments

The authors would like to express thanks to the VIRTUALTIMES consortium members for their helpful feedback and suggestions throughout the research project.

## Author Contributions

**Conceptualization:** Ville Johannes Harjunen, Niklas Ravaja.

**Data curation:** Ville Johannes Harjunen.

**Formal analysis:** Ville Johannes Harjunen.

**Funding acquisition:** Niklas Ravaja.

**Investigation:** Ville Johannes Harjunen.

**Methodology:** Ville Johannes Harjunen, Michiel Spapé.

**Visualization:** Ville Johannes Harjunen.

**Writing – original draft:** Ville Johannes Harjunen, Michiel Spapé, Niklas Ravaja.

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
