## [Decision Letter · Decision Letter 0]

17 Apr 2024

PONE-D-23-37645Anticipation of sexually arousing visual event leads to overestimation of elapsed timePLOS ONE

Dear Dr. Harjunen,

Thank you for submitting your manuscript to PLOS ONE. After careful consideration, we feel that it has merit but does not fully meet PLOS ONE’s publication criteria as it currently stands. Therefore, we invite you to submit a revised version of the manuscript that addresses the points raised during the review process.

We look forward to receiving your revised manuscript.

Kind regards,

Yansong Li

Academic Editor

PLOS ONE

Journal Requirements:

 [This work was supported by the VIRTUALTIMES (Horizon 2020, Contract No 824128; Authors funded by the project: V.J.H.; M.S.; N.R.) funded as a part of the Future and Emerging Technologies (FET) Project (FETPROACT-01-2018: Emerging paradigms and communities) of European Union.].  

 [This work was supported by the VIRTUALTIMES (Horizon 2020, Contract No 824128; Authors funded by the project: V.J.H.; M.S.; N.R.) funded as a part of the Future and Emerging Technologies (FET) Project (FETPROACT-01-2018: Emerging paradigms and communities) of European Union.].

  [This work was supported by the VIRTUALTIMES (Horizon 2020, Contract No 824128; Authors funded by the project: V.J.H.; M.S.; N.R.) funded as a part of the Future and Emerging Technologies (FET) Project (FETPROACT-01-2018: Emerging paradigms and communities) of European Union.].

Reviewers' comments:

Reviewer's Responses to Questions

**Comments to the Author**

1. Is the manuscript technically sound, and do the data support the conclusions?

Reviewer #1: No

Reviewer #2: Yes

2. Has the statistical analysis been performed appropriately and rigorously? 

Reviewer #1: No

Reviewer #2: Yes

3. Have the authors made all data underlying the findings in their manuscript fully available?

Reviewer #1: Yes

Reviewer #2: Yes

4. Is the manuscript presented in an intelligible fashion and written in standard English?

Reviewer #1: Yes

Reviewer #2: Yes

5. Review Comments to the Author

Reviewer #1: This is an interesting question. Also, the research is well-designed - it answers the research question. The inclusion of a condition in which a grey bar follows makes sense. Problems arise in terms of the data pre-treatment and analyses. Specifically, this is not a pre-registstred study and the critical p-values all hover around .05. Combine this with non-typical trial level exclusion (RTs 3 times beyond the MAD) criterion and person level exclusion (box plot) and we have a finding that might not reflect reality. The way forward is to pre-register a replication, consider effect sizes and uncertainty levels, and increase the sample size. By considering effect sizes and uncertainty levels we move away from the temptation to remove data points that make “p>.05”

Figure 3 indicates a rightward shift in the BP for the Anticipation + Blank condition. Is there something wrong here? - “As can be seen, the average BP for the anticipation+blank (M = 1509.89, 15 357 SE = 34.40) was lower than the average BP for the neutral condition (M = 1568.341, SE = 34.40), t(82) = 358 2.35, p = .021, d = .26.

The model needs to include random slopes otherwise, I need help to think of a reason for using MLM rather than fitting curves for each person in each condition and then using ANOVA.

The authors should reference and discuss: Angrilli A, Cherubini P, Pavese A, Manfredini S. The influence of affective factors on time perception. Percept Psychophys. 1997;39:972–982. Angrilli et al also used sexually arousing images even if they did not use a conditioning paradigm. Angrilli et al is a difficult paper to understand nonetheless, I think, some of the findings are relevant.

Reviewer #2: I would first like to thank the editors of PLOS ONE for the opportunity to review the manuscript entitled "Anticipation of sexually arousing visual event leads to overestimation of elapsed time" and congratulate the authors for their efforts. I think this study is interesting and great work.

I would like to make some suggestions to the authors to improve the quality of the manuscript. However, I also have some questions regarding this study.

Question:

1. One of my biggest concerns is the sample size of this study. I would like to know more about how the author computes the required sample size for this study design.

2. Considering the sample size was 42 individuals consisting of 29 females and 15 males if the difference between gender affects the result.

3. For the training session, after the training task, were there any tasks or assessments to investigate the correct ratio participants can distinguish between the short and long-lasting duration?

Suggestions:

For the “Introduction” session

1. I think the paragraph regarding threatening events (Line 56 - Line 73) can be shortened.

For the “Procedure” session

1. The training and bisection task sessions are not easy to follow. I suggest revising this part to be easy to understand for readers unfamiliar with this task.

2. In the bisection task session, for example, in line 202, I am trying to figure out what the appealing image indicates. Please verify. So I suggest unifying the words for the different stimuli in different places of the text, for example, erotic images and grey bars.

6. PLOS authors have the option to publish the peer review history of their article (what does this mean?). If published, this will include your full peer review and any attached files.

Reviewer #1: No

Reviewer #2: No

---

## [Author Response · Author response to Decision Letter 0]

29 May 2024

Response to reviewers

First, we would like to thank the reviewers for their thorough work, constructive comments, and suggestions. In response, we have made several changes including a thorough revision of the method section, clarification of the usage of MLM, and clarification of our outlier exclusion criteria, and correction to the Figure 3 in the results. We are convinced that these changes have improved the manuscript and hope they adequately reflect the feedback we received from the reviewers.

In the following two sections, we answer each reviewer individually specifying the changes made in the manuscript. We recommend reading the response letter attached in this submission as a separate file to see the graphs attached to the response.

Reviewer #1:

Comment 1: This is an interesting question. Also, the research is well-designed - it answers the research question. The inclusion of a condition in which a grey bar follows makes sense. Problems arise in terms of the data pre-treatment and analyses. Specifically, this is not a pre-registstred study and the critical p-values all hover around .05. Combine this with non-typical trial level exclusion (RTs 3 times beyond the MAD) criterion and person level exclusion (box plot) and we have a finding that might not reflect reality. The way forward is to pre-register a replication, consider effect sizes and uncertainty levels, and increase the sample size. By considering effect sizes and uncertainty levels we move away from the temptation to remove data points that make “p>.05”

Response to comment 1: Thank you for your comment. It is true that some of the p-values of the focal F-tests remain quite close to the .05 threshold (p = .030, p = .002, p = .026). However, these values are not the result of unjustified data exclusion or analysis. As we show below, the test results remain consistent even when using another method of trial-level outlier exclusion (z-score) which retains all but the most extreme outliers in the data or when using another statistical test (rmANOVA). In the future, pre-registered replication of these results is necessary to further ensure their robustness. As an initial examination of the effect of anticipation of appealing events on duration perception, the current study provides reliable evidence with adequate methodology for future research to build on. Below, we address the concerns raised by the reviewer more specifically.

First, what comes to data exclusion, the trial-level exclusion of RT outliers was based on median absolute deviation of RTs because it has been recommended over a commonly used Z-score method (Leys et al., 2013). It is true that detecting outliers by determining an interval spanning over the mean plus/minus three SDs remains a common practice. However, using z-score-based outlier removal is particularly problematic with RTs which tend to be strongly right-skewed with many extreme outliers. Z-scores are reliable if the distribution is normal bell curve-shaped but here the mean and the SD are influenced by the long tail of extreme values (Leys et al., 2013). This is why the MAD-based outlier detection and data exclusion was used instead. Indeed, the figure below shows that our RTs were clearly right-skewed and there were extreme outliers.

Leys C, Ley C, Klein O, Bernard P, Licata L. Detecting outliers: Do not use standard deviation around the mean, use absolute deviation around the median. Journal of experimental social psychology. 2013 Jul 1;49(4):764-6.

To further address the reviewer’s concern that we purposefully used a non-typical ourlier detection criterion for reaching statistical significance, we want to demonstrate that using a more commonly used Z-score-based outlier detection does not change the results. To this end, we conducted the analyses again using a more conventional exclusion method removing trials of which RT exceeded 3 SD from the mean. Applying this criterion instead of MAD = 3 criterion resulted in far less trials to be removed from the data (2.29% from the total trial count as compared to MAD-based exclusion of 9.12% reduction of trials). As the histogram of the RT values shown below reveals, some rather slow reactions were still remained in the data, which demonstrates that z-score-based outlier detection is suboptimal for our data. Nevertheless, the results did not change even if fitted the psychometric curves with the z-score-corrected data and comparing the means of condition-specific BPs. The main effect of cue condition was significant, F(2, 82) = 3.58, p = .033. Futhemore, the pattern of results remained the same with anticipantion+blank having smallest BP value (M= 1510, SE = 34.4) and neutral the largest BP (M = 1568 SE = 34.4) and the BP of anticipation+picture being in between the two (M = 1512, SE = 34.4). So, using a typical exclusion criterion and remaining almost all the outlier values in the data had no notable impact on the results. 

The reviewer also mentions person-level exclusion based on box plot observation. If we understand correctly, this critique concerns exclusion of a single individual’s observation with exceptionally low BP value in the anticipation + picture condition. This exclusion was done only for sensitivity analysis to test if the difference between neutral condition and anticipation + picture condition would remain significant after removing this outlier. The difference became non-significant after removing this outlier suggesting that the effect of anticipation + picture condition was influenced by this extreme value. However, no such influential extreme values were found from the other two conditions (anticipation + blank and neutral) and the follow-up analysis focusing on these two conditions was conducted using the same sample as before the sensitivity analysis. We hope that this information sufficiently addresses your concerns and brings clarity to the choices we have made. Finally, we have added the following text to the Method section to clarify why MAD-based outlier detection was used:

“MAD was used instead of commonly applied Z-score method due to the strongly right-skewed distribution of the RT values.”

Comment 2: Figure 3 indicates a rightward shift in the BP for the Anticipation + Blank condition. Is there something wrong here? - “As can be seen, the average BP for the anticipation+blank (M = 1509.89, 15 357 SE = 34.40) was lower than the average BP for the neutral condition (M = 1568.341, SE = 34.40), t(82) = 358 2.35, p = .021, d = .26.

Response to comment 2: Thank you for pointing out this error. We want to apologize for the confusion this flaw caused. Indeed, the legend labels were mixed in the script producing the Figure 3 and boxes and means for the anticipation+blank and neutral condition were reversed. This error has now been fixed. 

Comment 3: The model needs to include random slopes otherwise, I need help to think of a reason for using MLM rather than fitting curves for each person in each condition and then using ANOVA.

Response to comment 3: The models have only one categorical variable that was varied within participant that has been defined as random intercept. Adding the same condition variable both as random slope and random intercept does not work in this case because the random-effects parameters and the residual variance become unidentifiable. The reason for using MLM instead of repeated measures ANOVA was that in the later stage of analysis we include a person-level moderator and estimate cross-level interaction between condition and sexual arousal. Estimating such cross-level interaction is not possible with ANOVA/ANCOVA due to the assumption of homogenous regression slopes (Hoffman & Rovine, 2007). Therefore, using MLM is more appropriate than using ANOVA/ANCOVA in this case. We have now revised the Analysis section to provide this reason for using MLM instead of ANCOVA. Of course, it would have been possible to use repeated measures ANOVA to test the main effect of condition and then use MLM to test the cross-level interaction. Although the main effect of condition was significant also when using repeated measures ANOVA (F(1.48, 60.87) = 3.58, p < .05, ηp2 = .08), we opted for using MLM in both the main analysis and exploratory analyses as it made the analysis methods and models more consistent between the two phases (main and follow-up analyses). To address the concerns raised by the reviewer, the following excerpts were added to the text:

“No random slopes were included as adding the random slope of cue condition together with the random intercept of Subject id would have made the random-effects parameters and the residual variance to become unidentifiable.”

“Testing such cross-level interactions between repeated measures factors and person-level covariates with repeated measures ANCOVA violates the model’s assumption of homogenous regression slopes (Hoffman & Rovine, 2007). Therefore, MLM, which allows regression slopes to vary between conditions, was used for the statistical testing.”

Comment 4: The authors should reference and discuss: Angrilli A, Cherubini P, Pavese A, Manfredini S. The influence of affective factors on time perception. Percept Psychophys. 1997;39:972–982. Angrilli et al also used sexually arousing images even if they did not use a conditioning paradigm. Angrilli et al is a difficult paper to understand nonetheless, I think, some of the findings are relevant.

Response to comment 4: We are aware that Angrilli et at. also used erotic IAPS stimuli and our previous draft also cited the work. Their study had many important differences to the current study that we did not explicate in our previous version. First, the study by Angrilli et al (1997) examined how direct exposure to emotional stimuli influences duration estimation whereas we studied how anticipation of high arousal positive stimuli influence duration estimation. Then, Angrilli et al. used different timing tasks (reproduction and evaluation) than did we (temporal bisection task). The analyses were done adding both tasks to the same model which makes interpretation of the results a little bit difficult because reproduction tasks should result in underproduction of time when exposed to high arousal emotional stimuli whereas temporal bisection task should show an opposite (overestimation) effect, at least based on the pacemaker-accumulator model. The verbal evaluation task in turn should probably produce overestimation as a result of increased pulse rate. In response to the comment, we now discuss shortly their findings and methodological differences in the discussion:

“However, it is important to note that most of the previous studies used non-erotic positive stimuli. An early study by Angrilli and colleagues (1997), which used the same erotic IAPS stimuli as did we, found that direct exposure to erotic pictures resulted in an underestimation of elapsed time. While the current finding seems opposite to the findings by Angrilli and colleagues (1997), the discrepancy can well be due to different timing tasks used in the two studies (reproduction vs. temporal bisection task). Indeed, emotional conditions that produce overestimation in temporal bisection task may not similarly influence temporal reproduction (Gil & Droit-Volet, 2011). Nevertheless, our findings are in alignment with previous research related to temporal overestimation caused by anticipation of negative emotional events (Fayolle et al., 2015; Harjunen et al., 2021; Droit-Volet et al., 2010). As these studies utilized the same timing task and operated on the same sub-second time scale, their results can be considered more informative for understanding the current findings. In these studies on threat anticipation, the observed overestimation effect was attributed to an increase in arousal that was suggested to accelerate an internal pacemaker leading to higher number of pulses being accumulated over a time (Droit‐Volet et al., 2004). The current findings suggest that similar acceleration of pulse rate occurs also when expecting an appealing sexual event.”

Reviewer #2: 

I would first like to thank the editors of PLOS ONE for the opportunity to review the manuscript entitled "Anticipation of sexually arousing visual event leads to overestimation of elapsed time" and congratulate the authors for their efforts. I think this study is interesting and great work.

I would like to make some suggestions to the authors to improve the quality of the manuscript. However, I also have some questions regarding this study.

Comment 1: One of my biggest concerns is the sample size of this study. I would like to know more about how the author computes the required sample size for this study design.

Response to comment 1: Thank you for raising this important question. Indeed, we forget to report the justification of our sample size. The target sample size was decided based on our previous study and other studies which examined how anticipation of aversive stimuli influenced duration estimation. As these studies indicated the effect varying between medium (d = .76; Sarigiannidis et al., 2020; Harjunen et al., 2021) and large (d = 1.89; Fayolle et al., 2015) effect, we expected an effect of d = .75. Our power calculation indicated that detectin an effect of d = .75 with 80% statistical power and 0.05 alpha level would require a minimal sample of 39 participants. Therefore, we aimed at collecting a sample of over 40 individuals. We have added this information to the revised main text to justify our samples size.

“Required sample was estimated based on expected difference in averages BP values between emotional anticipation and neutral condition. Previous studies on anticipation of aversive events showed effect sizes varying between medium (d = .76; Sarigiannidis et al., 2020; Harjunen et al., 2021) and large (d = 1.89; Fayolle et al., 2015). Power analysis based on expected effect of d = .75 (https://jakewestfall.shiny apps.io/pangea/) informed that finding such effect with 80% statistical power and 0.05 alpha level required a minimal sample of 39 participants. Therefore, we aimed at collecting a sample of 40 or more individuals.”

Furthermore, we have expanded the part elaborating our statistical power to examine cross-level interaction between the cue condition and person-level arousal ratings:

“Finally, it is important to note that the sample size was underpowered for a reliable detection of small person-situation interaction effects. The requires sample size was estimated based on the main effect of cue condition that was the focus of our main analysis. Therefore, the observed small-to-medium (d = .37) interaction effect between cue condition and arousal may have been too small to be detected reliably with 80% statistical power when increasing the alpha level to the conventional 5% threshold. Therefore, further replication of the results is needed to ensure robustness of the reported findings.” 

Comment 2: Considering the sample size was 42 individuals consisting of 29 females and 15 males if the difference between gender affects the result.

Response to comment 2: Thank you for the comment. It is true that there might be differenced among genders in how their temporal estimation is influenced by such anticipatory circumstances. However, our sample size does not allow reliable comparisons between males and females. Especially the sample of males is quite small to give solid estimates. It is therefore possible that gender differences exist in temporal estimation during anticipation of sexually arousing stimuli, but we cannot address the question with the current data.

Comment 3. For the training session, after the training task, were there any tasks or assessments to investigate the correct ratio participants can distinguish between the short and long-lasting duration?

Response to comment 3: Thank you for the question. No, there was no other point at which we would have estimated their ability distinguish between long and short durations as in the training session. However, from the main data you can see that the extremely short (800 ms) and extremely long (2600) durations matching the anchor durations used in the training were classified correctly as short and long in

---

## [Decision Letter · Decision Letter 1]

21 Jun 2024

PONE-D-23-37645R1Anticipation of sexually arousing visual event leads to overestimation of elapsed timePLOS ONE

Dear Dr. Harjunen,

Thank you for submitting your manuscript to PLOS ONE. After careful consideration, we feel that it has merit but does not fully meet PLOS ONE’s publication criteria as it currently stands. Therefore, we invite you to submit a revised version of the manuscript that addresses the points raised during the review process.

We look forward to receiving your revised manuscript.

Kind regards,

Yansong Li

Academic Editor

PLOS ONE

Journal Requirements:

Additional Editor Comments:

1) it would be useful to add a sentence in the Discussion, mentioning the important role of anticipation of salient events in sexually dysfunctional patients. (e.g., citing the reference: Altered reward processing in patients with lifelong premature ejaculation. Scientific Reports.)

2) Please choose a public repository to host your data, such as the Open Science Framework (https://osf.io) or Zenodo (https://zenodo.org). This way, the data remains accessible to interested researchers.

Reviewers' comments:

Reviewer's Responses to Questions

**Comments to the Author**

1. If the authors have adequately addressed your comments raised in a previous round of review and you feel that this manuscript is now acceptable for publication, you may indicate that here to bypass the “Comments to the Author” section, enter your conflict of interest statement in the “Confidential to Editor” section, and submit your "Accept" recommendation.

Reviewer #2: All comments have been addressed

2. Is the manuscript technically sound, and do the data support the conclusions?

Reviewer #2: Yes

3. Has the statistical analysis been performed appropriately and rigorously? 

Reviewer #2: Yes

4. Have the authors made all data underlying the findings in their manuscript fully available?

Reviewer #2: Yes

5. Is the manuscript presented in an intelligible fashion and written in standard English?

Reviewer #2: Yes

6. Review Comments to the Author

Reviewer #2: The author has addressed my comments raised in a previous round of review and I feel that this manuscript is now acceptable for publication.

7. PLOS authors have the option to publish the peer review history of their article (what does this mean?). If published, this will include your full peer review and any attached files.

Reviewer #2: No

---

## [Author Response · Author response to Decision Letter 1]

25 Jun 2024

Rebuttal letter 

First, we would like to thank the reviewers and the editor for their thorough work, constructive comments, and suggestions. In response to the previous comments, we have incorporated new literature to the discussion and checked the references for retracted papers or papers that have been cited in the text but were not listed in the References section.

In the following, we the remaining questions and comments.

Journal Requirements:

Response: We have now gone through the references for missing or rectracted papers. No retracted papers were found. There were only two papers: one with erratum (error in figure labels) and one with a commentary. The references were formatted to PloS one- style and numbered citations were used in the text.

Additional Editor Comments:

1) it would be useful to add a sentence in the Discussion, mentioning the important role of anticipation of salient events in sexually dysfunctional patients. (e.g., citing the reference: Altered reward processing in patients with lifelong premature ejaculation. Scientific Reports.)

Response: Thank you for the suggestion. We have now added the following passage to the discussion section to take into account the relevance of our findings for sexual dysfunctions.

“Previous research on sexual dysfunctions, such as premature ejaculation (PE) disorder, has shown that men with inability to delay ejaculation during intercourses how also faster reactions when anticipating erotic images and reduced hedonic valuation of non-sexual rewards [48]. The current findings imply that the greater hedonic valuation of sexual rewards results also to stronger temporal dilation. Whether the temporal distortions caused by anticipated sexual rewards are stronger in people with sexual dysfunctions and addictions should be investigated in the future.”

2) Please choose a public repository to host your data, such as the Open Science Framework (https://osf.io) or Zenodo (https://zenodo.org). This way, the data remains accessible to interested researchers.

Response: the data has been openly available at Open Science framework. The link to the repository is provided in the data availability statement.

As there were no further comments from the reviewers, we would like to thank the Editor and the reviewers for their work on this manuscript that greatly improved in the process. We hope that the manuscript now meets the PLOS ONE’s publication criteria.

---

## [Editor Report · Decision Letter 2]

26 Jun 2024

Anticipation of sexually arousing visual event leads to overestimation of elapsed time

PONE-D-23-37645R2

Dear Dr. Harjunen,

We’re pleased to inform you that your manuscript has been judged scientifically suitable for publication and will be formally accepted for publication once it meets all outstanding technical requirements.

Kind regards,

Yansong Li

Academic Editor

PLOS ONE

---

## [Editor Report · Acceptance letter]

4 Jul 2024

PONE-D-23-37645R2 

PLOS ONE

Dear Dr. Harjunen, 

I'm pleased to inform you that your manuscript has been deemed suitable for publication in PLOS ONE. Congratulations! Your manuscript is now being handed over to our production team.

Kind regards, 

on behalf of

Dr. Yansong Li 

Academic Editor

PLOS ONE